# Characterization of a Novel Phage ΦAb1656-2 and Its Endolysin with Higher Antimicrobial Activity against Multidrug-Resistant *Acinetobacter baumannii*

**DOI:** 10.3390/v13091848

**Published:** 2021-09-16

**Authors:** Kyeongmin Kim, Md Maidul Islam, Dooyoung Kim, Sung Ho Yun, Jungmin Kim, Je Chul Lee, Minsang Shin

**Affiliations:** 1Department of Microbiology, School of Medicine, Kyungpook National University, 680 Gukchaebosang-ro, Jung-gu, Daegu 41944, Korea; rudals0691@knu.ac.kr (K.K.); islammm@knu.ac.kr (M.M.I.); dtwin@knu.ac.kr (D.K.); minkim@knu.ac.kr (J.K.); leejc@knu.ac.kr (J.C.L.); 2Bio-Chemical Analysis Team, Korea Basic Science Institute,162 Yeongudanji-ro, Ochang-eup, Cheongwon-gu, Cheongju-si 28119, Korea; sungho@kbsi.re.kr

**Keywords:** bacteriophage, genome sequencing, endolysin, antimicrobial activity, *Acinetobacter baumannii*

## Abstract

*Acinetobacter baumannii* is a nosocomial pathogen, which is a problem worldwide due to the emergence of a difficult-to-treat multidrug-resistant *A. baumannii* (MDRAB). Endolysins are hydrolytic enzymes produced by a bacteriophage that can be used as a potential therapeutic agent for multidrug-resistant bacterial infection in replacing antibiotics. Here, we isolated a novel bacteriophage through prophage induction using mitomycin C from clinical *A. baumannii* 1656-2. Morphologically, ΦAb1656-2 was identified as a Siphoviridae family bacteriophage, which can infect MDRAB. The whole genome of ΦAb1656-2 was sequenced, and it showed that it is 50.9 kb with a G + C content of 38.6% and 68 putative open reading frames (ORFs). A novel endolysin named AbEndolysin with an *N*-acetylmuramidase-containing catalytic domain was identified, expressed, and purified from ΦAb1656-2. Recombinant AbEndolysin showed significant antibacterial activity against MDRAB clinical strains without any outer membrane permeabilizer. These results suggest that AbEndolysin could represent a potential antimicrobial agent for treating MDRAB clinical isolates.

## 1. Introduction

*Acinetobacter baumannii* is an opportunistic ESKAPE (*Enterococcus faecium*, *Staphylococcus aureus*, *Klebsiella pneumoniae*, *Acinetobacter baumannii*, *Pseudomonas aeruginosa*, and *Enterobacter* spp.) pathogen responsible for nosocomial and community-acquired infections [1]. *A. baumannii* nosocomial infections can cause pneumonia, wound infection, urinary tract infection, and bacteremia in immunocompromised patients [2]. The biofilm formation ability of *A. baumannii* favors its survival in harsh environments, increased virulence, and multidrug-resistance capacity, leading to persistent chronic infection [3]. Multidrug-resistant (MDR) *A. baumannii* strains are most widespread in hospitals worldwide. The World Health Organization (WHO) categorized it as a top-priority pathogen for which new antimicrobial agents are urgently needed [4,5]. Besides searching for new antibiotics, bacteriophages could serve as a new therapeutic approach [6,7].

Bacteriophages (phages) are a diversified group of viruses that infect their host bacteria [8]. Research on bacteriophage therapy was renewed recently due to the emergence of the antibiotic efficacy crisis, although its discovery and use started about a century ago [9]. Phages have diverse structures, such as tailed, filamentous, or cubic, and most have double-stranded DNA (dsDNA) [10]. Therefore, phages are considered a potential therapeutic target because of their high specificity, safety, and rapidly modifiable properties [6]. Additionally, phages possess the capacity to infect the host bacteria, degrade the extracellular matrix, and hydrolyze biofilm [11].

Endolysins (lysins) are enzymatic protein products of bacteriophages produced at the end of the lytic cycle (end-stage phage replication) inside phage-infected bacterial cells [12]. Endolysin possesses either a single catalytic domain with amidase, peptidase, or glycosidase activity or another domain named the substrate-binding domain [13]. Endolysins can degrade the bacterial peptidoglycan layer when their cell-wall-binding domains specify substrate–enzyme recognition that favors the catalytic domains to lyse the peptidoglycan layer through bond cleavage [14]. Peptidoglycan protects bacteria from osmotic lysis, and thus, after peptidoglycan is degraded, bacteria cannot withstand osmotic pressure, causing lysis. The Gram-negative bacterial peptidoglycan layer is covered with an outer membrane, which reduces its susceptibility to lysins [15]. Lysin with outer membrane permeabilizers (OMPs), for example, EDTA, citric acid, and Triton X-100, can improve antimicrobial activity in Gram-negative bacteria [16,17,18]. However, a limited number of studies on OMP-independent endolysin activity against Gram-negative bacteria are available [19,20].

This study isolated a novel bacteriophage from clinical *A. baumannii* 1656-2 strain [21] using mitomycin C, which was then characterized phenotypically and genotypically. Moreover, complete genome analysis of ΦAb1656-2 to identify several functional open reading frames (ORFs) and the endolysin gene, AbEndolysin, was performed. We also purified AbEndolysin and tested its lytic potential against MDR *A. baumannii* strains. Our results show that AbEndolysin has high antimicrobial activity against MDR *A. baumannii* clinical strains.

## 2. Materials and Methods

### 2.1. Bacterial Strains and Culture Conditions

*Acinetobacter baumannii* ATCC 17978 was obtained from the American Type Culture Collection (ATCC), and clinical isolates were obtained from Kyungpook National University Hospital National Culture Collection for Pathogens (KNUH-NCCP). All bacterial strains were grown in Luria–Bertani, brain–heart infusion (BHI), or Mueller–Hinton broth or agar media. Bacterial strains were grown at 37 °C or 18 °C with antibiotics when required.

### 2.2. Bacteriophage Induction, Isolation, Purification, and Propagation

Phage induction, isolation, and purification from *A. baumannii* 1656-2 were performed as previously described by [22] with some modifications. Briefly, bacteria were grown at 37 °C until OD_600_ reached 0.9. Mitomycin C was then added to the culture media to a final concentration of 1 μg/mL and cultured overnight. The bacterial culture was centrifuged at 5000 rpm for 10 min to collect the supernatant, and chloroform was added to a concentration of 2% (*v*/*v*). The solution was centrifuged at 6000 rpm for 10 min, and the supernatant was filtered using a 0.22 µM filter (Millipore Corporation, Bedford, MA, USA). To confirm bacteriophage in the supernatant, 100 μL of supernatant and overnight-cultured *A. baumannii* ATCC 17978 (100 μL) were mixed with 0.75% warmed top soft agar, poured on BHI agar plates, and incubated at 37 °C for 18 h. To obtain a single bacteriophage, a single plaque was added in 200 μL of SM buffer (50 mM Tris-HCl (pH 7.5), 100 mm NaCl, 8 mm MgSO_4_, and 0.01% gelatin) and centrifuged at 13,500 rpm for 10 min to collect the supernatant. Propagation of Φ1656-2 phage was performed with *A. baumannii* ATCC 17978. The supernatant was used to infect the *A. baumannii* ATCC 17978, and bacteriophage was obtained by the above method. The phage was designated as ΦAb1656-2 and deposited in (we are awaiting a response from NCBI).

### 2.3. Bacteriophage Precipitation and Transmission Electron Microscopy

To precipitate the phage, the supernatant was mixed with 20% PEG 8000 and 2.5 M NaCl solution at a ratio of 1:4 and incubated in ice for 1 h. The mixture was then centrifuged at 10,000 rpm for 20 min, and the supernatants were discarded. The pellet was resuspended in SM buffer and stored at 4 °C. Purified bacteriophage and infected bacteria (MOI 1:100) were placed on a carbon-coated copper grid and negatively stained with 2% (*w*/*v*) uranyl acetate. Grids were observed in Hitachi HT 7700 Bio (Tokyo, Japan) Transmission Electron Microscope.

### 2.4. Phage DNA Extraction and Genome Sequencing

The Φ1656-2 phage DNA was isolated from precipitated phage using a Phage DNA Isolation Kit (Norgen Biotek, Schmon Pkwy, Thorold, ON, Canada) according to the manufacturer’s instructions. Whole-genome sequencing of Φ1656-2 phage DNA was performed using the HiSeq Xten platform through Macrogen (Seoul, Korea). Comparison and sequence similarity analysis was performed using the NCBI database (http://www.ncbi.nlm.nih.gov, accessed on 16 June 2021). ORFs were identified using the NCBI database. Three regions in the genome of Φ1656-2 phage were selected for PCR using both *A. baumannii* 1656-2 chromosomal DNA and ΦAb1656-2 genomic DNA as a template. The primers used in this PCR are listed in Appendix A. The whole-genome sequence data were deposited in GenBank under the accession number MZ675741.

### 2.5. Construction of pB4:AbEndolysin

The full length of the AbEndolysin gene was amplified by polymerase chain reaction (PCR) from Φ1656-2 genomic DNA with a forward primer 5′-CGGGGGCGGTGGTGGCGGCATGCCGCCTTCGGGCGG-3′ and reverse primer 5′-GTTCTTCTCCTTTGCGCCCTATATAACAACTCGATTGGCGATC-3′ designed for ligation-independent cloning [23]. The PCR product was treated with T4 DNA polymerase (New England Biolabs, Beverly, MA, USA) and inserted into a vector pB4, derived from the pET21a plasmid (Novagen, Darmstadt, Germany) [24]. The construct was confirmed by DNA sequencing analysis.

### 2.6. Purification of the Recombinant AbEndolysin

The pB4–AbEndolysin strain was transformed into *E. coli* BL21 (DE3) and grown at 37 °C, 150 rpm supplemented with ampicillin (100 μg/mL). When bacterial density reached OD600 ~0.5, IPTG was added at 150 mM final concentration to induce protein synthesis and incubated at 18 °C overnight. Bacterial cells were harvested by centrifugation at 6000 rpm for 15 min at 4 °C (rotor Beckman JA-14, Brea, CA, USA). After the breakdown of the cell wall by sonication, the supernatant was filtered by passage through a 0.45 µm filter (Millipore Corporation, Bedford, MA, USA) to remove cell debris. Then, the supernatant was applied to the Ni-NTA column (GE Healthcare, Little Chalfont, UK) and chromatographically purified using an FPLC chromatography system (Pharmacia, Stockholm, Sweden) [25]. To cleave between His-tag fused maltose-binding protein and AbEndolysin, TEV protease was treated in a dialysis buffer (20 mm Tris pH 7.5, 500 mm NaCl, 10 mm β-mercaptoethanol, 1 mm EDTA) at 4 °C overnight. After dialysis, the sample was reapplied to the Ni-NTA column and fractions were collected using the FPLC system and concentrated by Amicon Ultra-4 Centrifugal Filters Ultracel-10K (Millipore Corporation, Bedford, MA, USA). The purity of proteins was checked on 10% Tricine SDS–PAGE gel, concentration was measured using a NanoDrop 2000 Spectrophotometer (Thermo Fisher Scientific, Waltham, MA, USA), and they were then stored in the deep freezer.

### 2.7. Antibacterial Activity Test of AbEndolysin

AbEndolysin with different divalent ions (Zn^2+^, Ca^2+^, and Mg^2+^) at different concentrations were used to find appropriate cofactor and concentration for antibacterial activity against *A. baumannii*. The antimicrobial activity of AbEndolysin against selected *A. baumannii* clinical isolates under five sequence types, ST191 (KBN10P04948, KBN10P06070), ST208 (KBN10P04320, KBN10P04969), ST229 (KBN10P04598, KBN10P06126), ST369 (KBN10P04596, KBN10P04621), ST4514 (KBN10P05231, KBN10P05986), and ST784 (KBN10P04697, KBN10P05713), were determined using the microdilution method in 96-well round-bottom microplates [26]. Freshly cultured bacterial strains were suspended in MHB broth to finally produce 10^5^ CFU in each well. AbEndolysin with dialysis buffer (20 mm Tris pH 7.5, 200 mm NaCl) was added at 2-fold serially diluted concentration starting from 125 μg/mL until 0.195 μg/mL in each well. The wells were finally set up with 200 μL of 1× MHB supplemented with ZnCl_2_ (1 mM/mL). Fresh MHB was used as a negative control. The microtiter plate was incubated at 37 °C for 16 h, and the minimum inhibitory concentration (MIC) of AbEndolysin against tested strains was determined. Using the same protocol, antibacterial activity was determined against Gram-negative ESKAPE pathogens (*Enterococcus faecium* (clinical strain), *Staphylococcus aureus* ATCC 33591, *Klebsiella pneumoniae* ATCC 13883, *Pseudomons aeruginosa* ATCC 27853, and *Enterobacter cloacae* (clinical strain)).

## 3. Results

### 3.1. Isolation of ΦAb1656-2 from A. baumannii 1656-2

MDR clinical isolate *Acinetobacter baumannii* 1656-2 can significantly form biofilms that adhere to epithelial cells [21]. Biofilms protect microorganisms from the adverse environment and confer resistance to antibiotics, leading to the emergence of multidrug resistance. To find an alternative treatment option for MDR *A. baumannii* infection, we investigated new bacteriophages from *A. baumannii* 1656-2. To isolate bacteriophages from host bacteria, the host chromosomal DNA must be damaged. Mitomycin C covalently forms cross-links with DNA, causing DNA damage. Therefore, we treated early-stationary-phase bacterial cultures with mitomycin C and isolated bacteriophages (Figure 1A). To confirm whether the bacteriophage was isolated, a plaque assay was conducted using the host strain *A. baumannii* 1656-2 and standard strain *A. baumannii* ATCC 17978. Isolated phages could only infect *A. baumannii* ATCC 17978 and formed a plaque. To obtain highly purified bacteriophages, a single plaque was reinfected with *A. baumannii* ATCC 17978, and the presence of bacteriophage was confirmed through a repeated plaque assay (Figure 1). The isolated bacteriophage, ΦAb1656-2, forms small, clear, round plaques on *A. baumannii* ATCC 17978 lawn (Figure 1B). These results show that prophage in the clinical *A. baumannii* strain chromosome could be isolated using mitomycin C treatment and that the isolated bacteriophage can infect other *A. baumannii* strains.

### 3.2. Characterization of ΦAb1656-2 and Phage–Bacteria Interaction

To identify the morphology of isolated bacteriophages, we performed transmission electron microscopy (TEM) analysis. Carbon-coated grids with bacteriophages were negatively stained with uranyl acetate (2%) for TEM observation. There are three kinds of morphological classification of bacteriophages. Our TEM results show that ΦAb1656-2 belongs to the Siphoviridae family, according to the latest classification by the International Committee on the Taxonomy of Viruses (Figure 2A). We calculated the overall length and head diameter of ΦAb1656-2. The approximate length was 325.2 ± 13.3 nm; particularly, the phage head length was 69.3 ± 4 nm, the head diameter was 56.3 ± 4.8 nm, and the tail was 255.9 ± 9.3 nm (Figure 2B). To visualize the attachment of the bacteriophage on the surface of *A. baumannii* ATCC 17978, the purified ΦAb1656-2 and *A. baumannii* ATCC 17978 were mixed for 10 min at room temperature. TEM images show ΦAb1656-2 attached on the surface of *A. baumannii* ATCC 17978 (Figure 2C).

### 3.3. Whole-Genome Sequencing and Annotation

The genome of ΦAb1656-2 is integrated into the chromosome of *A. baumannii* 1656-2 The whole sequence of *A. baumannii* 1656-2 is already known [27] but the exact location and sequence of the ΦAb1656-2 sequence was unknown. Thus, we isolated genomic DNA to confirm the whole genomic sequence of ΦAb1656-2 (Figure 3A). The whole genome of the isolated genomic DNA was analyzed by next-generation sequencing. The analyzed whole-genome DNA showed the DNA molecule of 50,929 bp with an average G + C content of 38.6%. Based on BLAST-P, the genome predicted 68 putative ORFs; specifically, 25 putative hypothetical proteins, 14 putative functional proteins, and 29 phage-related proteins (Figure 3B). The analyzed ORF list is shown in Appendix A. From the results, we identified the AbEndolysin protein sequence. To confirm that the sequence of the ΦAb1656-2 identified through the whole-genome sequencing exists in the host *A. baumannii* 1656-2 chromosome, we selected three regions of the ΦAb1656-2 genome and amplified them by PCR. Three regions were amplified using *A. baumannii* 1656-2 chromosomal DNA and ΦAb1656-2 genomic DNA as templates. The PCR results show that the amplified products exhibit the same band size (Appendix A). These results indicate that the genomic sequence of ΦAb1656-2 exists in the chromosome of *A. baumannii* 1656-2.

### 3.4. Overexpression and Purification of Recombinant Endolysin

The putative endolysin protein (AbEndolysin) consisted of 195 amino acids (22 kD), and its pI value was 7.68 (http://web.expasy.org/compute_pi/, accessed on 14 May 2020). The predicted 3D structure of putative endolysin by the Phyre program showed that it comprises nine alpha helixes and three beta sheets (Figure 4A). Thus, the putative endolysin consists of an enzymatic active domain and cell-wall-binding domain. Furthermore, the AbEndolysin showed the N-terminal *N*-acetylmuramidase domain, i.e., the function is a degradation of peptidoglycan, and the C-terminal domain is the peptidoglycan-binding domain (Figure 4B). These predicted results show that AbEndolysin has a potential function in bacterial cell wall degradation. To test the antibacterial activity against MDR *A. baumannii*, we overexpressed and purified recombinant endolysin using FPLC (AKTA system). The purity of recombinant endolysin was confirmed by 12% Tricine SDS–PAGE gel (Figure 4C), and its concentration was measured using Nanodrop 2000.

### 3.5. Antibacterial Activity of AbEndolysin

To determine whether the recombinant endolysin has antibacterial activity against *A. baumannii* clinical strains, antimicrobial activity tests were conducted using the microdilution method. Endolysin requires a cofactor of divalent ions for amidase activity. Therefore, to determine which divalent ions are required for the antimicrobial activity of this recombinant endolysin, we conducted a cofactor identification experiment. Three cofactors were tested by treating Zn^2+^, Ca^2+^, and Mg^2+^ at concentrations of 10 mM, 1 mM, 0.1 mM, and 0 mM, respectively, wherein only Zn^2+^ exhibited antibacterial activity (Figure 5A). These results show that zinc is required as a cofactor for the antimicrobial activity of endolysin. In addition, in order to confirm whether the antibacterial effect by divalent cations exists, an antibacterial activity experiment was performed (Appendix A). These results show that 10 mM zinc has an antibacterial effect, whereas 1 mM zinc has no antibacterial effect. Therefore, 1 mM zinc was fixed to use in future antimicrobial activity tests.

Antibacterial activity assay was performed using 10 *A. baumannii* clinical isolates under five sequence types (ST-191, ST-208, ST-229, ST-369, ST-451, and ST-784) using the microdilution method to confirm that the recombinant endolysin has antibacterial activity. The endolysin was added at various concentrations (125 μg/mL until 0.195 μg/mL) with 200 μL of MHB per well. Treatment with endolysin inhibited the growth of almost all tested strains (Figure 5B). The MIC for all tested strains was at least 25 μg/well, except for two strains under ST-229 and ST-784. The difference in endolysin activity may be due to the outer membrane structure of these clinical strains. We also performed an antibacterial activity test with AbEndolysin against ESKAPE pathogens and it showed growth was inhibited only in *Staphylococcus aureus* ATCC 33591, a Gram-positive bacterium (Appendix A). These results suggest that AbEndolysin has specific antibacterial activity against the MDR-*A. baumannii* clinical strain, whereas it does not show antibacterial activity against other ESKAPE pathogens, except *Staphylococcus aureus*.

## 4. Discussion

The Gram-negative pathogen, *A. baumannii*, is responsible for healthcare-associated infections, and its intrinsic drug-resistance capacity led to the MDR organism [28]. To cure MDR pathogen-mediated infection, new antimicrobial agents or alternative antibiotics are urgently needed. In search of an alternative to antibiotics, bacteriophages could be a potential choice because of their self-replication, host specificity, and low toxicity [29]. The use of phase-encoded proteins, such as endolysin, as an alternative to antibiotics has already been established [30]. The main prospect of this study is to characterize a novel phase (ΦAb1656-2) and its endolysin (AbEndolysin) from a clinical *A. baumannii* strain, which has lytic activity against MDR *A. baumannii*.

*A. baumannii* harbors several inducible prophages in its genome because of its polylysogenic nature [31]. This study isolated and purified a novel *A. baumannii* phage (ΦAb1656-2) from MDR *A. baumannii* 1656-2 found in a university hospital in South Korea. Notably, this strain carries the *bla*_PER-1_ gene, having biofilm-forming and adherence properties [21]. Prophase induction using mitomycin C treatment was also previously reported [32]. The phage can inhibit the growth of MDR *A. baumannii.* The phage ΦAb1656-2 formed small, round plaques around *A. baumannii* 17,978 lawn (Figure 1); these results suggest that *A. baumannii* 17,978 is an appropriate indicator strain for this phage.

TEM showed that the phage ΦAb1656-2 belongs to the Siphoviridae family, similar to the other Siphoviridae *A. baumannii* phage AB1801 [33,34]. Like a typical Siphoviridae member, it has a non-contractile tail of 255 ± 9.3 nm, an icosahedral head with a diameter of 56.3 ± 4.8 nm, and a total length of about 325.2 ± 13.3 nm (Figure 2) [35]. Our TEM results show that ΦAb1656-2 could target MDR *A. baumannii* (Figure 2C). Furthermore, the lytic activity test showed that the phage is an MDR *A. baumannii*-specific lytic phage, which could be an alternative to antibiotics against MDR *A. baumannii.* Genome analysis showed that ΦAb1656-2 was 50,929 bp with an average G + C content of 38.6%. The genome predicted 68 putative ORFs; specifically, 25 putative hypothetical proteins, 14 putative functional proteins, and 29 phage-related proteins (Figure 3B). We also confirmed that the sequence of ΦAb1656-2 was laid in the chromosome of *A. baumannii* 1656-2 by PCR (Appendix A).

We identified a sequence for endolysin in the genome of ΦAb1656-2 and named it AbEndolysin. We performed multiple sequence alignment of AbEndolysin with 14 different *Acinetobacter* phage endolysins to compare the sequence homology. Since most of the Gram-negative bacterial lysins have only a catalytic domain, we only used catalytic domain sequences from all 15 lysins including AbEndolysin during alignment. Our alignment results show that there is high sequence homology among other *Acinetobacter* endolysins, but very few sequences are similar to AbEndolysin (Appendix A).

Bioinformatic analysis suggested that AbEndolysin had both catalytic and cell-wall-binding domains (Figure 4B). The catalytic domain is *N*-acetylmuramidase, which can cleave bonds between *N*-acetylmuramic acid and *N*-acetylglucosamine, causing degradation of the peptidoglycan layer. *Pseudomonas aeruginosa* bacteriophage endolysin ΦKZ and EL also have two domains, although the location of the catalytic domain is opposite compared to AbEndolysin [36]. We also tested the requirement of divalent metal ions for the maximal lytic activity of AbEndolysin. Among three metal ions (Zn^2+^, Ca^2+^, and Mg^2+^), only Zn^2+^ was shown to be required for AbEndolysin lytic activity (Figure 5A). LysB4, a phage endolysin from *Bacillus cereus,* also required the same divalent metal ions for lytic activity [37]. We also determined the optimal concentration of Zn^2+^ for the activity of AbEndolysin, as this divalent ion itself has antibacterial activity (Appendix A). Several *A. baumannii* endolysins (LysAB2, PlyF307, LysABP-01, PlyAB1, ABgp46, and LysAB54) are used to treat Gram-negative bacterial infection in vitro and in vivo, which has been reviewed and reported in [19]. Among them, PlyF307, LysAB21, and LysAB54 have OMP-independent antibacterial activity against *A. baumannii* [20,37,38,39]. Several other endolysins, such as LysSAP26, LysSS, and Abtn-4, have a broad range of antimicrobial activity against Gram-positive and Gram-negative pathogens [40,41]. We used five sequence-type (ST-191, ST-208, ST-229, ST-369, ST-451, and ST-784) *A. baumannii* strains found in Kyungpook National University Chilgok Hospital to test the enzyme activity of AbEndolysin. It had high antibacterial activity without any OMPs against all tested sequence-type strains (Figure 5B). Our antibacterial activity test with AbEndolysin using ESKAPE pathogens showed antibacterial activity only against *Staphylococcus aureus* ATCC 33591, a Gram-positive bacterium (Appendix A).

In conclusion, this study isolated a novel *A. baumannii* bacteriophage, named ΦAb1656-2, from a clinical strain and demonstrated its morphological and genomic characteristics. We also purified AbEndolysin from ΦAb1656-2, which has antibacterial activity against MDR *A. baumannii* clinical strains and *Staphylococcus aureus* ATCC 33591, a Gram-positive bacterium. AbEndolysin has antibacterial activity without OMPs that make it an essential candidate for treating MDR bacterial infection.

## Figures and Tables

**Figure 1 viruses-13-01848-f001:**
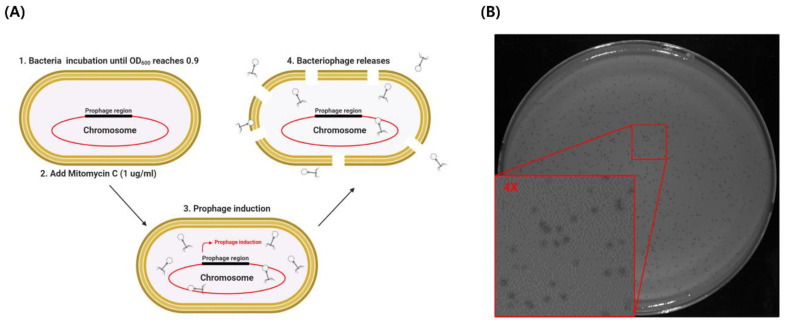
Lysogenic phage isolation from *A. baumannii* 1656-2 clinical strain. (**A**) Schematic of prophage isolation from *A. baumannii* 1656-2 clinical strain with the treatment of mitomycin C. (**B**) Plaque assay of isolated ΦAb1656-2 with *A. baumannii* ATCC 17978 on a BHI agar plate.

**Figure 2 viruses-13-01848-f002:**
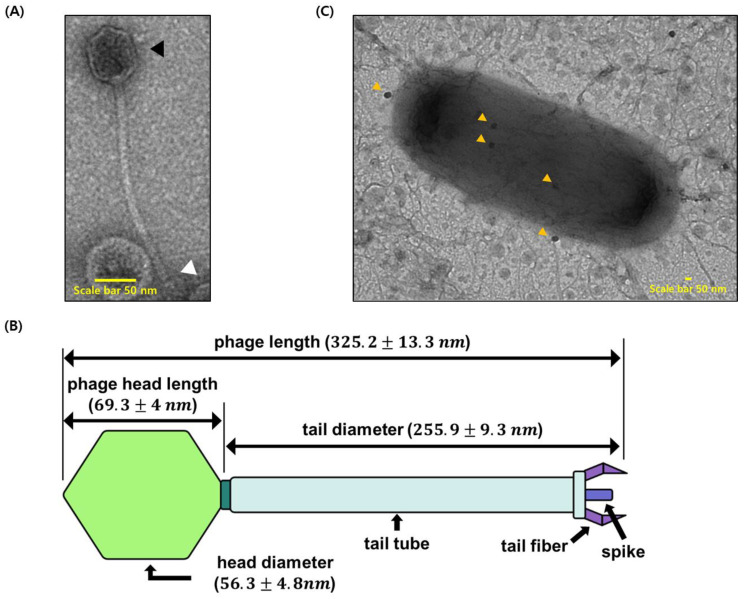
TEM images of phage ΦAb1656-2. ΦAb1656-2 was observed by TEM after being negatively stained with 2% uranyl acetate. (**A**) The morphology of ΦAb1656-2 showed that it is under the Siphoviridae family. The black arrowhead indicates the phage head, and the white arrowhead indicates the phage tail. (**B**) Illustration showing the shape and dimensions of ΦAb1656-2. Phage length and diameter were measured from the TEM images. (**C**) ΦAb1656-2 and host *A. baumannii* ATCC 17978 interaction by TEM. Orange arrowhead indicates ΦAb1656-2. The scale bar is 50 nm.

**Figure 3 viruses-13-01848-f003:**
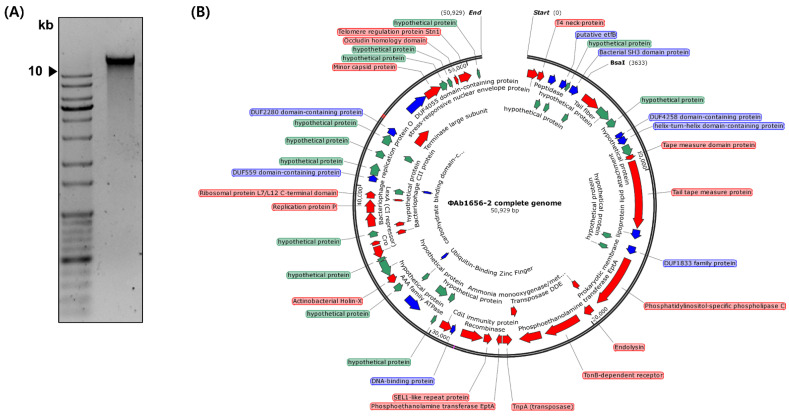
Complete genome of ΦAb1656-2. (**A**) Isolated genomic DNA of ΦAb1656-2 loaded on 1% agarose gel. The black arrowhead indicates 10 kb. (**B**) Genome map of ΦAb1656-2. The genome size of ΦAb1656-2 was identified as 50.9 kb, and 68 putative ORFs were predicted. Green represents predicted hypothetical proteins, blue represents putative functional proteins, and red represents phage-related proteins.

**Figure 4 viruses-13-01848-f004:**
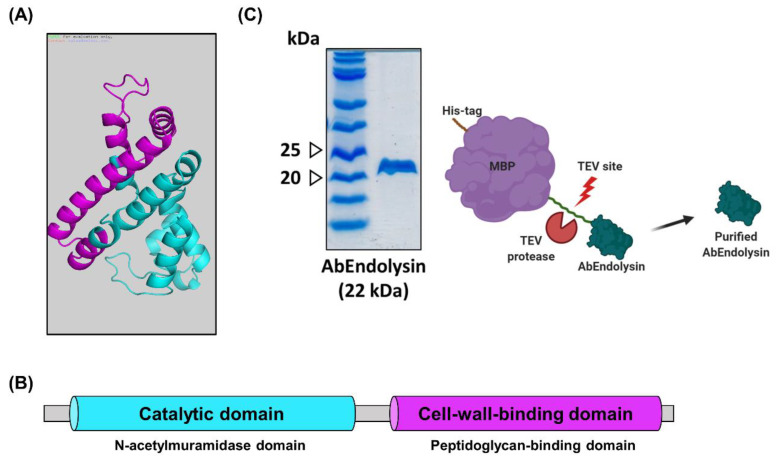
Purification of recombinant AbEndolysin. (**A**) The 3D structure of AbEndolysin was predicted using Phyre. It is composed of nine alpha helixes and three beta sheets. (**B**) AbEndolysin consists of two domains. The N-terminal is the *N*-acetylmuramidase domain (sky-blue), and the C-terminal is the peptidoglycan-binding domain (purple). (**C**) Schematic of AbEndolysin. Purified recombinant AbEndolysin was separated on a 10% Tricine SDS–PAGE gel.

**Figure 5 viruses-13-01848-f005:**
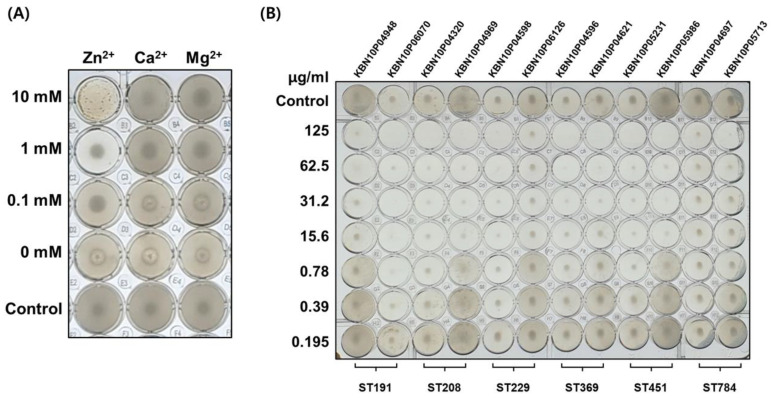
Antibacterial activity of AbEndolysin. (**A**) Identification of cofactor of AbEndolysin. The concentration of AbEndolysin was fixed, and the divalent ions Zn^2+^, Ca^2+^, and Mg^2+^ were treated at concentrations of 10 mM, 1 mM, 0.1 mM, and 0 mM, respectively. (**B**) Antimicrobial activity of AbEndolysin against MDR-*A. baumannii* clinical strains (ST-191, ST-208, ST-229, ST-369, ST-451, and ST-784). AbEndolysin was serially diluted two-fold from the highest concentration of 125 μg/mL to confirm its inhibitory concentration. Fresh bacterial culture in MHB was used as control.

## Data Availability

The findings of this study are available within this paper and its Appendix A file. The whole-genome sequence data discussed in this publication have been deposited in the NCBI’s gene expression omnibus database (http://www.ncbi.nlm.nih.gov/geo/, accessed on 2 August 2021) and can be accessed using the accession number MZ675741.

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
