# Peer review of "Characterization of a Novel Phage ΦAb1656-2 and Its Endolysin with Higher Antimicrobial Activity against Multidrug-Resistant Acinetobacter baumannii"

_viruses, 2021, doi:10.3390/v13091848_

Round 1

Reviewer 1 Report

The authors reported the isolation of Acinetobacter phage and its characterization of endolysin. Their results are very interesting, but I think the authors can add more informative results very easily. I would like to ask the author to add some discussion as follows.

1) Figure 1B is not clear. Please prepare an easy-to-see photo.

2) Please prove by PCR that the phage genome is inserted in the genome of the lysogen. I also expect that the location where the phage genome is inserted in the lysogen genome can be easily identified.

3) Many endolysins as shown in Fig. 4 have been reported. I think it is easy to discuss sequence homology using a phylogenetic tree with those endolysins and the Acinetobacter phage endolysin. The authors can also discuss the three-dimensional structure and catalytic residues from the reported endolysin papers.

Author Response

Please see the attached file (including rebuttal letter, revised manuscript, supplementary data)

Reviewer 2 Report

 This study reports isolating a novel bacteriophage through prophage induction using mitomycin C from a clinical A. baumannii and expressing the lysin Abendolysin, which is active to A. baumannii without outer membrane permeabilizers (OMPs). In view that there are already many lysins reported for Gram-negative bacteria, the current results are too preliminary to judge its significance for publication. The major concerns are:

  1. On the bacteriophage purification, which host bacteria was used for the purification? Since the phage was obtained by induction, it is a temperate phage. It is strange to see that the phage could form plaques during purification. Please provide more details on the phage purification and provide some explanation on this phenomena.
  2. The sequence of AbEndolysin must be provided and compared its similarity with other lysins reported for A. baumannii.
  3. Fig. 4: It is strange to see that the lysin has a cell wall binding domain. Normally lysins for gram negative bacteria just have a catalytic domain. More experiments are needed to confirm this conclusion, for example, by expressing truncated proteins containing only the CBD or the Catalytic domain.
  4. There are more lysins reported showing activity against A. baumannii without OMPs (For example, Khan FM, Gondil VS, Li C, Jiang M, Li J, Yu J, Wei H and Yang H (2021) A Novel Acinetobacter baumannii Bacteriophage Endolysin LysAB54 With High Antibacterial Activity Against Multiple GramNegative Microbes. Front. Cell. Infect. Microbiol. 11:637313. doi: 10.3389/fcimb.2021.63731). The main problem facing these lysins is that they could not work in complicated matrix, such as, sera and milk. Therefore, it will be interesting to see the activity of AbEdolysin in different matrix, including human sera and milk etc, as well as, buffers with different pH and ionic strength.
  5.  More data are also needed for testing the activity of the lysin against other gram-negative bacteria besides A. baumannii.

Author Response

(The authors gave the same response as above.)

Reviewer 3 Report

The presented manuscript refers to phage endolysin belonging to ΦAb1656-2 phage. The report is important and valuable, however, in this description, there is a lack of comparative analyses of the known endolysins. The antimicrobial activity is also questionable. The title suggests the high activity against bacteria. Does 25ug of the enzyme per 100 bacterial cells suggest the high antimicrobial activity? When can it be judged there is high activity of the enzyme? There is also a lack the discussion about Zn 2+ usage, while these ions are antibacterial themselves. In general, since there are more and more reports about phage endolysins, there is a need to make a comparative analysis among them, including showing the novelty and advantage.  I recommend improving this manuscript by adding the comparative analysis and convince the Reader why the enzyme contains high activity.

Author Response

(The authors gave the same response as above.)

Round 2

Reviewer 1 Report

I think the manuscript has been properly revised.

So, I agree to publish this article.

Author Response

Thank you for your comments.

Reviewer 2 Report

In the revised manuscript, the authors have answered some of the previous concerns. However, it still needs some clarification:

  1. When comparing the sequence of ABEdolysin, what is the criteria for you to choose  other 13 lysins? Some recently reported lysins (such as LysAB54) were not included.
  2. Since most of the lysins for gram-negative bacteria do not contain cell wall binding domain, please use only the catalytic domain sequence when doing the comparison (Fig. S4).
  3. The authors mentioned another reference (Yves Briers et.al.;2007, doi.org/10.1111/j.1365-2958.2007.05870.x) to prove that some lysins have two domains. But why cite another unrelated reference (ref No 34, line 316) in the manuscript? Furthermore, the cell binding domain of lysin KZ144 is at N-terminal. While the cell binding domain of ABEdolysin is at C-terminal, which is usually the place for lysins against Gram-positive bacteria. The binding experiment as in Yves Briers et.al.;2007 must be done to confirm this. Especially, since ABEndolysin seems having activity against S. aureus, the binding of the cell binding domain to S. aureus must be done also.

Author Response

Please see the attached file (rebuttal, revised manuscript including supplementary data).

Reviewer 3 Report

The Authors covered all the requirements that were suggested in the 1st round

Author Response

Thank you for your comment.